# Advancements in Battery Monitoring: Harnessing Fiber Grating Sensors for Enhanced Performance and Reliability

**DOI:** 10.3390/s24072057

**Published:** 2024-03-23

**Authors:** Kaimin Yu, Wen Chen, Dingrong Deng, Qihui Wu, Jianzhong Hao

**Affiliations:** 1School of Marine Equipment and Mechanical Engineering, Jimei University, Xiamen 361021, China; 202212855006@jmu.edu.cn (K.Y.); drdeng@jmu.edu.cn (D.D.); qihui.wu@jmu.edu.cn (Q.W.); 2School of Ocean Information Engineering, Jimei University, Xiamen 361021, China; 3Institute for Infocomm Research (I^2^R), Agency for Science, Technology and Research (A★STAR), Singapore 138632, Singapore

**Keywords:** fiber Bragg grating sensors, battery sensor, safety monitoring, battery

## Abstract

Batteries play a crucial role as energy storage devices across various industries. However, achieving high performance often comes at the cost of safety. Continuous monitoring is essential to ensure the safety and reliability of batteries. This paper investigates the advancements in battery monitoring technology, focusing on fiber Bragg gratings (FBGs). By examining the factors contributing to battery degradation and the principles of FBGs, this study discusses key aspects of FBG sensing, including mounting locations, monitoring targets, and their correlation with optical signals. While current FBG battery sensing can achieve high measurement accuracies for temperature (0.1 °C), strain (0.1 με), pressure (0.14 bar), and refractive index (6 × 10^−5^ RIU), with corresponding sensitivities of 40 pm/°C, 2.2 pm/με, −0.3 pm/bar, and −18 nm/RIU, respectively, accurately assessing battery health in real time remains a challenge. Traditional methods struggle to provide real-time and precise evaluations by analyzing the microstructure of battery materials or physical phenomena during chemical reactions. Therefore, by summarizing the current state of FBG battery sensing research, it is evident that monitoring battery material properties (e.g., refractive index and gas properties) through FBGs offers a promising solution for real-time and accurate battery health assessment. This paper also delves into the obstacles of battery monitoring, such as standardizing the FBG encapsulation process, decoupling multiple parameters, and controlling costs. Ultimately, the paper highlights the potential of FBG monitoring technology in driving advancements in battery development.

## 1. Introduction

Batteries have revolutionized various industries by offering high energy density, long lifespan, and recharge ability [1,2,3]. Their widespread use in electric vehicles, energy storage systems, and mobile technologies has accelerated the global energy transition [4,5,6,7,8]. However, the complex chemical reactions in batteries pose safety risks such as thermal runaway and capacity degradation [9,10]. Even routine charging and discharging induce simultaneous changes in micro-chemical properties and macro-physical parameters, including temperature, stress, gas by-products, and electrolyte solubility. External factors such as collisions, extrusions, and environmental changes can further impact battery performance and safety [11,12]. Battery runaway phenomena typically initiate within the battery and extend throughout the entire battery, eventually reaching the battery pack. Despite extensive efforts to improve battery materials and design, these challenges persist. Consequently, the research focus has shifted towards monitoring detailed chemical changes and assessing the health states within the battery to enhance battery properties, predict service life, and provide early warnings for safety risks [13]. Table 1 outlines sensors for monitoring various parameters, along with their installation locations, advantages, and disadvantages. According to the measured battery parameter, monitoring methods can be categorized into:

(1) Detecting the microstructure or spectral properties of the battery material: Traditional methods like X-ray diffraction (XRD), scanning transmission electron microscope (STEM), Raman spectroscopy, infrared (IR) spectroscopy, nuclear magnetic resonance (NMR), and mass spectrometry enable the direct observation of the crystal structure and microscopic surface morphology of electrodes or the study of the molecular composition of liquid electrolytes. However, these methods require sample preparation and testing in the laboratory using expensive and sophisticated instruments. While accurate, they are not real-time.

(2) Detecting physical phenomena accompanying chemical reactions in batteries: The real-time monitoring of physical phenomena associated with chemical reactions (temperature changes, strains, and deformations) is typically accomplished using load cell, strain gauge, thermocouple, or IR thermography. While these methods provide real-time insight into chemical reaction processes, they have limitations in accurately depicting changes in battery characteristics and health. Thermocouples are typically placed on the surface of the cell and have limitations in directly assessing internal temperatures. Monitoring large area battery pack temperatures is challenging due to point measurements and wire conductivity issues. IR thermography, while useful, is limited by resolution and accuracy, making it difficult to accurately determine surface and internal temperatures. Load cell and strain gauge methods can measure cell expansion and contraction, but are affected by temperature.

(3) Detecting chemical properties of battery substances: Analyzing the chemical properties of battery substances, such as acidity, alkalinity, and refractive index (RI), is commonly achieved using electrochemical workstations or electrochemical analyzers. Techniques like cyclic voltammetry (CV), constant current constant voltage (CCCV), and electrochemical impedance spectroscopy (EIS) explore ionic transport during battery chemical reactions. Mass spectrometry assesses solute and solvent concentrations in the electrolyte, along with its potential of hydrogen (pH) value. Cycle life testing simulates charging and discharging cycles to study electrode material changes and electrolyte degradation in real time. While this method addresses the limitations of the previous two types, providing direct monitoring of battery material properties, the comprehensive monitoring of each battery remains impractical.

Despite the wide variety of sensors used to monitor various parameters, current sensors are still unable to provide real-time and accurate monitoring of the quality, reliability, and life (QRL) of batteries. Fiber optic sensors are prominent in battery sensing due to their compactness, high sensitivity, chemical inertness, resistance to electromagnetic interference, and multiplexing capability. Distributed fiber optic sensing, while capable of covering the battery pack comprehensively, faces challenges such as inadequate spatial resolution, low measurement accuracy, and the requirement for manual calibration with precise positioning [14]. Point-type fiber optic sensors like the Mach–Zehnder (MZ) interferometer, Michelson interferometers, Sagnac interferometers, and Fabry-–Pérot (FP) interferometers can address these issues but may struggle with monitoring the battery pack at multiple points. Quasi-distributed fiber Bragg grating (FBG) sensors offer a solution to these challenges as they can be conveniently deployed and integrated on the surface or inside the cell, delivering more precise detection results [15,16,17]. Notably, FBG spectroscopy provides insights into ionic motion at the molecular level, making it the most widely used sensor in battery sensing today [18]. However, some recent fiber optic sensors have attempted to monitor changes in battery chemistries, such as Surface Plasmon Resonance (SPR) spectroscopy, but it is difficult to use them to quantitatively characterize changes, and they do not allow for large-scale applications [19,20,21,22,23].

This paper aims to further explore the accurate measurement of battery structure health status and remaining life using FBG sensors, based on a comprehensive investigation of various types of sensing systems and their monitoring parameters. The article begins by elaborating on the chemical reaction mechanisms and potential adverse reactions within the battery during operation. Subsequently, it delves into the current application status of FBG sensors in monitoring internal parameters of batteries. Finally, it scrutinizes the future prospects and challenges associated with the application of FBG sensors in battery monitoring.

**Table 1 sensors-24-02057-t001:** Various battery sensing methods.

Method	Parameters	Characteristics	Location
Fiber Bragg Grating (FBG) [24]	Temperature, Strain, RI, and more	Small size, immune to electromagnetic interference, quasi-distributed measurement, and real-time monitoring	Internal, External
Resistance Temperature Detector [25]	Temperature	High accuracy, wide linear range, and slow response	Internal, External
Thermistor [26]	Temperature	Small and cost-effective, fast response, and limited accuracy	Internal, External
Thermographic Imaging [27]	Temperature	Measures overall temperature distribution, limited resolution and accuracy	External
Isothermal Calorimetry [28]	Temperature	Complex equipment, poor real-time performance	External
Infrared (IR) Thermal Imaging [29]	Temperature	Graphic display of temperature distribution, real-time monitoring	External
Pressure Sensor [30]	Pressure	Real-time monitoring of gas pressure, larger size	External
Load Cell [31]	Strain	Requires support structure, internal parameters cannot be directly measured	External
Strain Gauge [32]	Strain	Used for material stress analysis, attached to battery surface	External
X-ray Diffraction (XRD) [33]	Strain	Material discrimination, phase transformation, structural variation, particle size distribution, and lattice size	External
X-ray Photoelectron Spectroscopy (XPS) [34]	Strain	Elemental component, valence variation, and energy shifts	External
Digital Image Correlation [35]	Strain	Acquires surface deformation information, cannot access internal information	External
IR Spectroscopy [36]	Electrolyte composition	Provides information about the structure of the molecule	External
Raman Spectroscopy [37]	Electrolyte composition	Measurement of molecular vibrational modes	External
Nuclear Magnetic Resonance (NMR) [38]	Electrolyte composition	Provides information about the structure of the molecule	External
Mass Spectrometry [39]	Electrolyte composition	Provides information on the mass and structure of the molecule	External
Optical Fiber Evanescent Wave [40]	Electrolyte composition	Monitors electrolyte composition, complex sample preparation	Internal
Machine Learning Algorithms [32]	SOC/SOH	Data-driven analysis, requires a large number of high-quality data	External
Scanning Transmission Electron Microscope (STEM) [41]	SoC/SoH	Morphological change, element distribution, compositional analysis, and structural change	External
Equivalent Circuit Model (ECM) [42]	SoC/SoH	Limited accuracy, cumulative model error	External

## 2. Battery Degradation Mechanism and Fiber Optic Monitoring Principle

### 2.1. Battery Principle

Deng et al. from our team designed a variety of special electrode materials and applied them to various new energy storage devices, such as lithium-ion (Li-ion) batteries, sodium-ion batteries, lithium-sulfur batteries, and supercapacitors, which greatly improved the performance of the batteries. In addition, they have systematically studied the interfacial reactions in energy storage systems and elaborated various reaction mechanisms [43,44,45,46,47]. While different battery types undergo unique charging and discharging processes, they fundamentally rely on internal chemical reactions for energy conversion. During the charging phase, an external power supply introduces electrons into the battery, merging them with positive ions within the energy storage material, transitioning the battery from a low-energy state to a high-energy state. Upon discharge, the chemical bonds in the energy storage material break, returning the battery to a low-energy state. This process releases stored electrons and positive ions, supplying energy to external devices. These processes induce alterations in microchemical properties and macroscopic physical parameters, including heat, stress, gas byproducts, and electrolyte solubility, impacting battery performance and cycle life. Currently, operational battery systems predominantly comprise 78% Li-ion batteries, with sodium and lead-acid batteries accounting for 11% and 4%, respectively. Other storage technologies, such as flow batteries, electrochemical capacitors, sodium-ion batteries, and nickel- and zinc-based batteries, collectively contribute to the remaining 7% [48]. Consequently, the majority of reported fiber optic sensing battery monitoring studies have focused on Li-ion batteries. This paper specifically centers on the online monitoring of Li-ion batteries.

#### 2.1.1. Anode Degradation

During the charging and discharging process of Li-ion batteries, several deleterious phenomena occurring at the anode significantly impact the QRL of the battery [49,50,51,52,53,54,55]. These phenomena are primarily observed in the following three aspects:

(1) The formation of a solid electrolyte interface (SEI) layer on the anode surface leads to a decline in conductivity and ion transport rate. When a chemical reaction occurs between the electrolyte and the anode, a SEI layer forms on the anode surface, accompanied by the generation of gases [56,57,58,59,60]. With an increasing number of charging and discharging cycles, the SEI layer thickens on the anode surface due to mechanisms such as the solvent molecules diffusion, cleavage, and side-reaction product accumulation. The SEI layer obstructs contact between the active material and the electrolyte, reducing the electrical conductivity and ion transport rate of the battery, ultimately degrading the QRL. Moreover, under harsh conditions such as high temperature, high current load, and electrode breakage, the growth of the SEI layer may accelerate, leading to a decrease in battery capacity [61,62,63].

(2) Internal stresses within the anode material can result in cracking or fracturing. The uneven embedding or dislodging of Li-ions in the electrode material, causing varying volume changes in different regions, has been observed [64,65]. This volume change induces uneven internal stresses in the electrode material, ultimately leading to electrode cracking and potential fracture. High current loads and extreme temperatures can expedite this phenomenon. As cracks widen on the electrode surface, the SEI layer grows on the newly exposed electrode surface, further amplifying the uneven stress distribution within the battery. Additionally, the reduction reaction generates gases, increasing pressure within the battery, potentially resulting in battery cracking.

(3) Lithium plating occurring on the anode surface poses the risk of causing internal short circuits. This phenomenon induces a series of associated side reactions, including the deposition of lithium metal and lithium dendrite growth. Li-ions undergo a reduction reaction on the anode surface to form lithium metal deposition [60]. Consequently, lithium metal no longer participates in redox reactions, leading to an irreversible loss of battery capacity. Furthermore, the deposited lithium metal extends from the anode surface, forming a tree-like structure known as lithium dendrites. The growth of lithium dendrites has the potential to breach the separator, causing an internal short circuit and triggering thermal runaway in the battery. The formation and evolution of the lithium anode coating are influenced by various factors, such as the electrolyte’s nature, the positive-to-negative electrode capacity ratio, the operating temperature, and the charging rate.

#### 2.1.2. Cathode Degradation

During battery operation, the cathode also undergoes changes detrimental to the QRL of the battery, such as loss of active material, cathode cracking, and cathode electrolyte interface (CEI) formation [66,67,68,69,70].

(1) The loss of active material from the cathode represents a prevalent aging process. Transition metals such as nickel, manganese, cobalt, and iron in the cathode may dissolve in the electrolyte [69], leading to a decline in battery capacity and performance. This phenomenon is particularly accelerated under conditions of high temperature and frequent charging and discharging cycles.

(2) Cathode cracking can occur due to charge insertion or extraction and gas generation. Specifically, the uneven insertion and extraction of lithium ions during charging and discharging can cause irreversible phase transitions in the cathode structure, resulting in internal stresses that crack the cathode [71]. This essentially reduces the amount of acceptable lithium ions in the cathode, resulting in diminished battery capacity. Additionally, gases generated within the battery contribute to cathode cracking; at elevated temperatures, metal oxides may lose oxygen, and under high pressures, the electrolyte may decompose, producing gases. The accumulation of these gases raises the internal air pressure within the battery, potentially leading to cathode cracking.

(3) Similar to the anodic SEI layer, the CEI layer is mainly generated by the chemical reaction between the cathode and the electrolyte. However, due to the higher cathode voltage, the volume of the CEI layer is considerably smaller than that of the SEI layer. Its formation results in a reduction in active materials, such as lithium, uneven pressure leading to alterations in the cathode structure, and even the development of cracks. Particularly under conditions of high voltage and high state of charge (SOC), these issues become more severe, significantly impacting the QRL of the battery.

### 2.2. Fiber Optic Sensing Principle

The FBG sensor is essentially an optical wavelength selector that utilizes periodic changes in RI along an optical fiber to diffract and reflect light. When external factors such as temperature, strain, or pressure impact the fiber grating, the periodic RI of the grating changes, leading to the reflection or diffraction of light at a specific wavelength. FBG sensors have a wide range of applications in fields such as structural health monitoring, oil and gas pipeline surveillance, aerospace, and the medical industry [18,72,73,74,75,76,77,78,79,80,81,82,83,84,85].

#### 2.2.1. Fiber Bragg Grating

Fiber gratings are formed by interferential irradiation of optical fibers by laser or ultraviolet light beams [86,87,88]. As shown in Figure 1, when broadband light is incident on a fiber grating sensor, specific wavelengths of light are reflected while others are transmitted.

The wavelength of the reflected light is so-called the Bragg wavelength, which is proportional to the period of the grating and the effective RI of the fiber. It can be expressed as:(1)$$\lambda_{B}=2n_{neff}\Lambda ,$$
where $n_{eff}$ denotes the effective RI of the optical fiber, $\lambda_{B}$ denotes the Bragg wavelength, and Λ denotes the grating period. When a FBG sensor is subjected to temperature and/or stress, its period and equivalent refraction change, resulting in a change in Bragg wavelength. The relationship between Bragg wavelength change and temperature/strain can be expressed by the following equation:(2)$$\frac{\Delta \lambda_{B}}{\lambda_{B}}=\left(\alpha +\beta \right)\Delta T+\left(1-\rho_{e}\right)\Delta \varepsilon ,$$
in which
(3)$$\;\alpha =\frac{\partial \Lambda /\Lambda }{\partial T},\beta =\frac{\partial n_{\mathrm{eff}}/n_{\mathrm{eff}}}{\partial T},\rho_{e}=\frac{n_{eff}^{2}}{2}\left[\left(1-\nu \right)p_{12}-\nu p_{11}\right],$$
where ΔT is the change of the FBG’s temperature, α denotes the thermal expansion coefficient, β denotes the thermo-optic coefficient, Δε is the FBG’s strain, $\rho_{e}$ denotes the effective optoelastic coefficient of the fiber core, ν denotes the Poisson’s ratio, and $p_{11}$ and $p_{12}$ denote the Pockel’s coefficients of the strain optical tensor [89]. Therefore, FBG sensors can be used to detect temperature and/or strain. The mainstream spectrum analysis device is shown in Figure 2. The system consists of a light source, a circulator, a spectral analysis device, a photodetector, and a data acquisition card. The light signal from the light source passes through the circulator and enters the optical fiber with the FBG sensor. The FBG sensor reflects the light back to the circulator and is spectrally decomposed. The photodetector measures the light intensity of different frequencies, which is then converted to 16-bit electrical signals by the data acquisition card. Finally, the signals are processed by a computer.

#### 2.2.2. Tilted Fiber Bragg Grating

Unlike conventional FBG, Tilted FBG (TFBG) features a specific angle Bragg grating tilted with respect to the optical axis of the fiber [90,91,92,93]. This tilt angle induces coupling between the forward-propagating fiber core modes and the cladding modes, resulting in the emergence of multiple closely spaced resonance peaks in the transmission spectrum of the sensor, as illustrated in Figure 3.

The resonant wavelengths of the core modes exhibit high sensitivity to a broad range of physical parameters (temperature, strain, pressure, etc.) but remain insensitive to changes in the external RI. On the other hand, the resonant wavelength of the cladding mode is highly sensitive to variations in the external RI. Consequently, the resonant wavelength of the cladding mode shifts in response to changes in the surrounding refractive index. The mode wavelengths $\lambda_{cl}\left(i\right)$ coupled to the cladding can be expressed as:(4)$$\lambda_{cl}\left(i\right)=\frac{\left(n_{eff,co}+n_{eff,cl}\left(i\right)\right)\Lambda }{\cos\theta },$$
where $\lambda_{cl}\left(i\right)$ is the wavelength of the ith cladding mode, and $n_{eff,co}$ and $n_{eff,cl}\left(i\right)$ are the effective RI of the fiber core mode and the ith cladding mode, respectively. θ is the tilted angle between the grating planes and the cross-section of the fiber.

## 3. Online Monitoring of Battery Degradation

FBG sensors have garnered significant attention in battery monitoring applications since their initial use in 2013 [94]. Originally employed to monitor the real-time surface temperature distribution, surface strain, and SOC of batteries [64,95,96,97,98,99,100,101,102], their monitoring capabilities have subsequently expanded to include temperature, strain, pressure, and chemical reaction processes within the battery.

### 3.1. Monitoring Chemical Formulae and Stereo Structures

Traditional methods for characterizing the microstructure and crystal structure of battery materials involved techniques such as STEM, Scanning Electron Microscopy (SEM), and XRD to analyze the loss of Li-ion and the extent of electrode fracture and cracking [41,103,104,105,106]. EIS was employed to evaluate the electrochemical impedance of the battery, providing insights into the growth of the SEI layer [107]. Compositional analysis of battery materials was performed using NMR, mass spectrometry, IR spectroscopy, and Raman spectroscopy, offering molecular-level insights into the state of electrodes and electrolytes [36,37,38,39].

Different fiber optic sensors enable a variety of spectroscopic analysis techniques, including SPR spectroscopy, absorption spectrum, IR spectroscopy, and Raman spectroscopy. These spectral analysis techniques are safe and rapid methods that have been widely used to monitor the chemical composition and evolution of various batteries.

Fiber optic evanescent wave sensors (FOEWS) are used to analyze the absorption spectra of the electrodes to monitor the changes in the material inside the battery. Its transmittance intensity is affected by the lithium content of the graphite anode, which is directly related to the SOC of the Li-ion battery. Therefore, FOEWS can be used to estimate the SOC of the battery. Further, a decrease in the amount of change in its transmittance indicates a decrease in the amount of change in the lithium content of the graphite, which implies that the capacity of the Li-ion battery is degraded. Therefore, the magnitude of FOEWS transmittance can also decode the state of health (SOH) of the battery [108].

Silicon-free background Raman spectroscopic measurements of various electrolyte components were conducted by incorporating a hollow-core fiber optic sensor into a Li-ion pouch battery. The aim was to undertake a thorough examination of the chemical degradation mechanism of the electrolyte. Spectroscopic analysis revealed changes in the ratio of carbonate solvents and electrolyte additives corresponding to variations in battery voltage. Moreover, the hollow-core fiber optic sensor showcased its potential for monitoring lithium ion dissolution dynamics [37]. Hollow-core fiber optics offer solutions to several limitations associated with silica fiber optic probes in monitoring nearby electrolyte compositions. These limitations include relatively weak Raman signals, short light-matter interaction lengths (largely confined to the fiber tip), and sensitivity issues induced by silica, while simultaneously eliminating silica-generated Raman signals.

Fiber optic IR spectroscopy has been employed to monitor the insertion and removal of Na-ions from electrodes, as well as the phase transitions experienced by materials inside the cell during cycling. Gervillié-Mouravieff et al. implemented operando IR fiber evanescent wave spectroscopy using sulfur-based (sulfide, selenide, and telluride) glass fibers with a transmission window in the range of 3 to 13 μm [109].

Localized electrochemical events at the electrode interface can be screened by monitoring changes in the SPR spectra of a TFBG sensor with a gold film. In 2022, Wang et al. [110] successfully quantified ion transport kinetics and electrolyte–electrode interactions at the electrode surface of a battery using this sensor, as shown in Figure 4.

The sensor, with its longer penetration depth and propagation length, allows for the real-time monitoring of electrochemical kinetics without interfering with the normal operation of the battery. Experiments conducted on a zinc-ion water battery illustrated the visualization of a two-step ion insertion mechanism at the MnO_2_ cathode during the discharge process, distinguishing the intercalation processes of H^+^ and Zn^2+^. Additionally, it was demonstrated that a thin layer of poly (3,4-ethylenedioxythiophene) coated on the surface of the electrodes improves battery capacity and cycle stability.

In 2024, Han et al. [111] successfully embedded a TFBG sensor onto the electrode surface of a lithium battery, enabling the monitoring of mass transfer kinetics and lithium dendrite growth at the anode nanoscale interface. In addition, it was demonstrated that the artificial SEI layer can reduce the substance concentration gradient on the anode surface, which helps to inhibit the generation of lithium dendrites, notably based on the ionic conductor artificial SEI layer.

Traditional techniques, while capable of accurately characterizing the internal state of a battery, require expensive and complex equipment that cannot be monitored in real time.

### 3.2. Monitoring Physical Phenomena of Batteries

The microscale reactions, such as lithium ion insertion/precipitation, volume expansion, gas generation, and dendrite formation, inside the electrode particles can be manifested in the macroscale as regular stress evolution, temperature changes, etc. The loss of battery materials and the insertion and detachment of ions can lead to changes in the internal stress of the battery and even lead to battery rupture. The internal battery strain can provide information about the change of battery thickness and the evolution of electrode stress [112]. The polarization and exotherm of chemical reactions result in internal temperatures significantly higher than external temperatures [113,114], offering a more reflective measure of the charging and discharging state of the battery.

In 2013, Yang et al. [94] conducted investigations on Li-ion button and cylindrical batteries, achieving real-time temperature monitoring by installing FBG sensors on the battery surface. Comparative experiments with thermocouple sensor monitoring demonstrated that FBG sensors can accurately track temperature changes in Li-ion batteries under normal and excessive charging and discharging conditions. They exhibited excellent thermal response performance during battery charging and discharging, providing a temperature resolution of 0.1 °C and a sensitivity of 10 pm/°C, making them a viable alternative to thermocouple sensors.

In 2016, Novais et al. [13] positioned two FBG sensors inside and outside Li-ion pouch cell to monitor their temperature changes, as shown in Figure 5.

The results revealed that internal FBG sensors recorded temperature variations of up to 2 °C between the two areas, while external FBG sensors exhibited temperature variations of only 0.2 °C. In the center and electrode areas of the cell, external temperature measurements showed minimal variations, approximately 1.5 ± 0.1 °C, whereas corresponding internal temperature variations in the center area of the cell were 4.0 ± 0.1 °C, and internal temperatures varied by 4.7 ± 0.1 °C. These temperature fluctuations were correlated with current magnitude, with the highest temperature peaks occurring at the end of charging and discharging. Additionally, the influence of strain on temperature measurements was negligible due to the pouch cell’s very small thickness.

In 2018, Amietszajew et al. [115] utilized an FBG sensor to evaluate the internal temperature of cylindrical Li-ion battery. The internal temperature increased by 5 °C compared to the external temperature. To eliminate the effect of strain on the temperature measurement, an aluminum loose sleeve was used to protect the FBG sensor encapsulated with a polyamide coating. In addition, fluorinated ethylene propylene (FEP) heat shrinkable tubing was used to prevent the aluminum tubing and polyamide coating from interfering with the electrolyte, as shown in Figure 6a.

The results showed that the battery could safely withstand a maximum charging current 6.7 times higher than the manufacturer’s specified value. As a result, a fast-charging protocol was developed to improve charging efficiency, reducing charging time by more than five times while ensuring the continued safe operation of the battery.

In the same year, Fleming et al. [116] of the same research group used four FBG sensors to monitor the internal temperature distribution of a cylindrical Li-ion battery. These sensors were threaded through a custom aluminum tube coated with FEP into the interior of the battery, as shown in Figure 6b. The results show that in the region close to the anode and cathode, the temperature difference from the surface of the battery is as high as 6 °C during discharge and 3 °C during charging. In addition, the corrosion resistance of the FBG sensor material to the electrolyte was demonstrated, as well as the long term stability of the FBG sensors during the battery cycling process.

In 2022, Liu et al. [117] employed a femtosecond laser FBG sensor to monitor the operating temperature inside a cylindrical Li-ion battery. The integration scheme of the FBG with a thermocouple sensor into the battery is depicted in Figure 7.

The FBG sensor and a thermocouple were inserted inside the battery, while another thermocouple was positioned outside the battery as a temperature reference. The results demonstrated that the FBG sensor and the thermocouple sensor exhibited similar temperature response curves during charge/discharge cycling at different rates (0.5 C, 1 C, and 2 C). It is noteworthy that the FBG sensor exhibited a smaller baseline change. The internal cell temperature detected by the FBG sensor was 0.96 °C higher than the external temperature during the 0.5 C cycle, 1.78 °C higher during the 1 C cycle, and 3.78 °C higher during the 2 C cycle. Additionally, the reflective wavelength of the FBG sensor showed minimal change over the 2 MPa internal pressure range, indicating that the internal pressure change had a negligible effect on temperature.

In 2023, Wu et al. [15] employed an FBG array film (FBGAF) for assessing the internal temperature of a hard-shelled Li-ion battery. The FBGAF integrates a flexible thin-film sensor with five FBG sensors, as illustrated in Figure 8. In experiments with pulse discharge rates of 30 C, the internal temperature increased by approximately 13–15 °C, reaching a maximum temperature of 50 °C. Concurrently, the external temperature increased by about 10 °C, reaching a maximum temperature of 45 °C. Notably, the FBGAF were able to resist corrosion of the electrolyte solution for a long period of time.

In 2016, Bae et al. [118] applied FBG sensors by pasting and implanting them on Li-ion battery anodes to monitor strain changes, as depicted in Figure 9. The pasted FBG sensors measured longitudinal strain, while the implanted FBG sensors measured both longitudinal and transverse strain. The results revealed that during charging, electrode expansion caused the peak wavelength to shift towards longer wavelengths, and during discharging, electrode contraction caused the peak to shift towards shorter wavelengths.

In 2016, Ganguli et al. [119] utilized two FBG sensors to measure internal strain inside a Li-ion pouch battery for estimating SOC and SOH. In static cycling tests with fixed charging and discharging rates, the estimated SOC error did not exceed 1%. During dynamic cycling tests involving different charge/discharge rates, the estimated SOC error remained below 2.5%. Furthermore, they developed a method to estimate SOH with an error of no more than 1.1%.

In 2022, Blanquer et al. [120] utilized FBGs and external force sensors to monitor stress changes inside and outside a Li-ion battery, respectively. Throughout battery cycling, changes in the fiber Bragg wavelength reflected stress variations and exhibited correlation with voltage. Importantly, in symmetric all-solid-state Li-ion batteries, external force sensors faced limitations in monitoring stress changes outside the battery, particularly those occurring in the electrodes. In contrast, FBG is not constrained by monitoring electrode stress due to its small size.

In 2022, Miao et al. [70] integrated FBG sensors into the positive electrode of a Li-ion pouch battery to monitor stress changes. Through an exploration of various sulfur embedding and releasing mechanisms, they discovered that stress changes are not only associated with volume alterations but also influenced by material properties. Specifically, under the solid–solid mechanism, the polyacrylonitrile (PANS) cathode material exhibited significant stress changes, whereas under the solid–liquid–solid mechanism, the ketjen black/sulfur (KB/S) cathode material displayed a smaller stress change.

In 2022, Unterkofler et al. [121] explored encapsulation methods for FBG sensors used within the Li-ion battery. Various encapsulation methods for strain relief were compared, including polyether ether ketone (PEEK) with a single-point adhesive, fused silica tubes with a polyimide coating, fused silica tubes without a polyimide coating, and fused silica tubes with a two-point adhesive but without a polyimide coating. The results indicated that the encapsulation method using fused silica tubes with polyimide coatings is user-friendly, provides effective fiber protection, has a smaller diameter, and exhibits rapid response to temperature changes.

In 2017, Fortier et al. [24] employed an FBG sensor to monitor the temperature and strain inside a Li-ion coin cell. To safeguard the positive electrode from damage, two isolation layers were added above and below the embedded FBG sensor, as illustrated in Figure 10a–d.

The results revealed a temperature difference of 10 °C between the inside and outside of the cell at a charge/discharge rate of C/20. The small size of the cell contributed to minimal strain changes inside the cell.

In 2019, Nascimento et al. [122] embedded hybrid sensors consisting of FBG sensors and FP cavities at the top, middle, and bottom of the inside of a Li-ion pouch battery to monitor strain and temperature changes, as shown in Figure 11.

The simultaneous measurement of strain and temperature changes can be expressed by the following equation:(5)$$\left[\begin{array}{c} \Delta \varepsilon \\ \Delta T \end{array}T\right]=\frac{\left[\begin{array}{cc} -K_{FP_{T}} & K_{FBG_{T}}\\ K_{FP_{\varepsilon }} & -K_{FBG_{\varepsilon }} \end{array}\right]}{K_{FP_{\varepsilon }}K_{FBG_{T}}-K_{FP_{T}}K_{FBG_{\varepsilon }}}\left[\begin{array}{c} \Delta \lambda_{FBG}\\ \Delta \lambda_{FP} \end{array}\right],$$
where $K_{FBG_{T}}$ and $K_{FBG_{\varepsilon }}$ represent the temperature and strain sensitivity coefficients of the FBG sensor, respectively. $\Delta \lambda_{FBG}$ states the wavelength shift of FBG sensors. Similarly, $K_{FP_{T}}$, $K_{FP_{\varepsilon }}$ represent the temperature and strain sensitivity coefficients of the FP cavity, respectively. $\Delta \lambda_{FP}$ represents the wavelength shift of the FP cavities. The results show that the FBG sensor mounted at the bottom of the battery is the most sensitive to strain changes at 65.0 ± 0.1 με, and the one in the middle is the most sensitive to temperature changes at 3.3 ± 0.1 °C.

In 2022, Xi et al. [112] positioned two FBG sensors within a Li-ion coin cell to meticulously observe temperature and stress variations. As shown in the Figure 12, one FBG sensor nested inside a quartz capillary was subjected to temperature only, while the other was subjected to both temperature and strain.

The findings revealed a maximum temperature alteration of 0.42 °C and a maximum strain change of 11.5 με. Notably, the strain variations exhibited greater magnitude during the initial two charging and discharging cycles and diminished in subsequent cycles. The existence of strain within the cell was substantiated by the observation of lithium dendrites through SEM at the conclusion of the charging and discharging processes.

In 2018, Lao et al. [123] utilized a TFBG sensor with an attached gold film to monitor the charging and discharging processes of supercapacitors. The results showed that the SPR spectral response was closely related to the charge stored in the supercapacitor. Notably, the SPR spectral response demonstrated a stable and distinct reflection of multiple charge/discharge cycles in supercapacitors.

The aforementioned alterations in physical parameters, such as temperature and strain induced by battery chemistry, may not precisely portray the authentic state of the battery. Temperature variations can be influenced by factors like battery design and operating environment, while strain may be subject to various elements, including material phase transitions and battery structure. Certain materials, such as lithium titanate and lithium iron phosphate, exhibit minimal or even negligible strains. Consequently, methodologies reliant solely on monitoring these physical parameters might not offer an accurate determination of the battery’s state. Fortunately, additional phenomena during battery operation, such as variations in electrolyte RI, pH fluctuations, and the generation of gases due to battery side reactions, present avenues for acquiring more precise and detailed insights into a battery’s QRL.

### 3.3. Monitoring Compounds and Gas Products

Accompanying substances in battery chemistry serve as key information for deciphering the internal operational status of the battery. High temperatures can prompt metal oxides to shed oxygen molecules, and elevated pressure can cause electrolytes to decompose, resulting in gas production. If the gas generation rate surpasses the diffusion rate, it may lead to battery cracking. Gas generation can also occur during the initial formation of the SEI layer. Simultaneously, redox reactions may alter the electrolyte’s solubility. FBG sensors prove instrumental in monitoring these gas byproducts and electrolyte concentrations, facilitating the analysis of SEI layer formation and ion transport kinetics.

In 2020, Huang et al. [124] utilized FBG sensors integrated into conventional single-mode fiber (SMF) or microstructure optical fiber (MOF) to simultaneously monitor temperature and gas pressure inside the battery, as shown in Figure 13a.

Two FBG sensors were positioned in a 0.8 mm diameter hole at the battery’s center to mitigate the impact of internal strain. The hole was subsequently sealed with epoxy resin, as shown in Figure 13b. The operando calorimetry method transforms temperature into quantifiable thermal events, enabling the tracking of SEI layer formation and battery life. Temperature and gas pressure can be decoupled using the following equation:(6)$$\begin{array}{c} \;\Delta P=\frac{k_{T,\mathrm{SMF}}\Delta \lambda_{B,\mathrm{MOF}}-k_{T,\mathrm{MOF}}\Delta \lambda_{B,\mathrm{SMF}}}{k_{T,\mathrm{SMF}}k_{P,\mathrm{MOF}}-k_{T,\mathrm{MOF}}k_{P,\mathrm{SMF}}},\\ \Delta T=\frac{\Delta \lambda_{B,\mathrm{SMF}}-k_{P,\mathrm{SMF}}\Delta P}{k_{T,\mathrm{SMF}}}, \end{array}$$
where $k_{P,\mathrm{SMF}}$ = −0.3 pm/bar and $k_{T,\mathrm{SMF}}$ = +10 pm/°C represent the pressure sensitivity and the temperature sensitivity of the SMF sensor, respectively. $k_{P,\mathrm{MOF}}$ = −2.7 pm/bar and $k_{T,\mathrm{MOF}}$ = +10 pm/°C represent the pressure sensitivity and the temperature sensitivity of the MOF sensor, respectively. $\lambda_{B,\mathrm{MOF}}$ and $\Delta \lambda_{B,\mathrm{SMF}}$ represent the wavelength shift of the MOF-FBG sensor and SMF-FBG sensor, respectively.

In 2021, Desai et al. [125] integrated an FBG sensor with a MOF to monitor the heat and pressure within a cell, aiming to optimize the electrolyte formulation for sodium-ion batteries. The study investigated the electrolyte degradation rate in three batteries utilizing the same electrolyte with the addition of various electrolyte additives. The three batteries included Na_3_V_2_(PO_4_)_2_F_3_ (NVPF)/hard carbon (HC) full-cell, NVPF/NVPF symmetric, and HC/HC symmetric batteries. The electrolyte composition comprised ethylene carbonate vinyl acetate (EC)/propylene carbonate (PC)/dimethyl carbonate (DMC). The electrolyte additives were formulated as 0.5 wt.% sodium oxalato (difluoro) borate (NaODFB), 3 wt.% vinylene carbonatevinyl (VC), 3 wt.% succinonitrile (SN), and 0.2 wt.% VC and 0.2 wt.% tris-trimethylsilylphosphite (TMSPi). The findings revealed that cells with additives, particularly in the NVPF electrodes, exhibited a lower rate of electrolyte degradation and considerably reduced heat and gas generation compared to cells without additives.

In 2023, Mei et al. [126] devised a fiber optic sensor comprising an FBG and an FP interferometer (FPI) to elucidate the thermal runaway mechanism and its progression in three distinct SOC cases (100% SOC, 50% SOC, and 0% SOC). Cells with 100% SOC exhibited thermal runaway internal temperatures reaching up to 510 °C, with a maximum temperature difference between the interior and surface exceeding 180 °C. The thermal runaway behavior of the 50% SOC cell closely resembled that of the 100% SOC cell, registering an internal temperature of 440 °C. Conversely, the 0% SOC cell displayed no thermal runaway, reaching a maximum internal temperature of 330 °C. The study demonstrated that the FBG-FPI exhibited exceptional resistance to high temperatures during the entire thermal runaway phase.

In 2019, Nedjalkov et al. [127] employed an innovative FBG sensor for monitoring the Li-ion pouch battery, incorporating an additional optical waveguide embedded in the fiber cladding. As illustrated in Figure 14, the first FBG sensor is integrated into the fiber core, while the second FBG sensor is integrated into the optical waveguide at the fiber’s edge.

This design enhances the sensor’s sensitivity to changes in the electrolyte solution’s RI. Experimental results demonstrate the sensor’s capability to detect subtle changes in electrolyte properties and accurately characterize battery capacity degradation. It has no significant effect on battery performance, although traces of pressure by the sensor are detected on the electrodes. Notably, the sensor exhibited early detection of thermal runaway in abuse tests.

In 2021, Huang et al. [90] utilized a TFBG sensor inserted into a hollow cylindrical Li-ion cell to decipher its internal temperature and electrolyte RI. Due to the TFBG’s immunity to the strain within the battery, its spectral changes are solely influenced by variations in temperature and RI. The wavelength interval between the resonance wavelength and the Bragg resonance provides RI information. Moreover, monitoring the amplitude peaks of the TFBG cladding mode enables the tracking of electrolyte turbidity, offering valuable insights into the formation process of the SEI layer.

Compared to the first two types of detection methods, monitoring the chemical reaction’s accompanying substances provides more accurate and real-time insights into the SOC, SOH, and Remaining Useful Life (RUL) of the battery. However, challenges persist in decoupling the electrolyte RI from other measured quantities, such as barometric pressure and stress; demodulating optical signals to extract the measured quantities; and effectively monitoring multi-battery configurations.

### 3.4. FBG Measurement Accuracy and Sensitivity

The monitoring of the internal parameters of the battery was achieved based on FBG sensors. Table 2 shows a summary of some FBG sensors used to monitor different parameters of the battery (temperature, strain, pressure, electrolyte RI, and SOC). Table 2 illustrates that the FBG sensing systems developed by Novais et al. [13], Nascimento et al. [122], Huang et al. [124], and Desai et al. [125] exhibit the highest temperature accuracy of 0.1 °C. Furthermore, Nascimento et al. [122]’s FBG sensing system demonstrates the highest temperature sensitivity at 40 pm/°C. Nascimento et al. [122] established the peak strain accuracy and sensitivity of the FBG sensor at 0.1 με and 2.2 pm/με, respectively. Huang et al. [124] reported the top pressure measurement accuracy and sensitivity of the FBG sensor as 0.14 bar and −0.3 pm/bar. The highest RI measurement accuracy and sensitivity for FBG sensors were determined to be 6 × 10^−5^ RIU and −18 nm/RIU, respectively, by Huang et al. [90]. While these studies achieve notable precision and sensitivity for individual parameters, simultaneous high accuracy and sensitivity for multi-parameter measurements remain challenging due to the complexities associated with multi-parameter decoupling.

## 4. Challenges and Outlooks

FBG sensing technology for battery applications is rapidly advancing from conventional laboratory measurements towards practical field monitoring applications. However, the future development of FBG sensors for internal battery monitoring encounters several challenges.

Firstly, the design and packaging of FBG sensors for internal battery applications confront complex technical requirements. The variable and intricate operating environment within the battery, coupled with the delicate nature of FBGs, necessitates stable sensor operation. Optimizing fiber material, structure, and surface metal plasma coating becomes imperative to withstand the harsh chemical environment within batteries [128,129], preventing electrolyte leakage, and improving the QRL of the battery [130]. Our fiber optic grating sensor design, manufacturing, and packaging technologies developed for the construction, aerospace, robotics, chemical, entertainment, biological, and pharmaceutical industries may be transferable to internal and external battery monitoring [131,132,133,134,135,136,137].

Secondly, accurately parsing FBG sensor data used for battery assessment is a major challenge [70,110,121,124,125]. (1) Due to the sensitivity of fiber optic sensors to multiple parameters, such as temperature and stress, achieving decoupling and precise measurement of multidimensional parameters is crucial for accurately evaluating the health status of batteries [138,139,140,141]. This includes addressing the coupling of temperature and strain, electrolyte RI, and other influencing factors, such as the solid electrolyte layer versus the electrochemical interfacial layers. (2) Effective signal processing techniques can significantly reduce the impact of noise on optical signals, thereby enhancing the accuracy of battery condition measurements [142,143,144,145,146,147,148]. For instance, the Activation Function Dynamic Averaging (AFDA) algorithm outperforms the Frequency Domain Dynamic Averaging (FDDA) algorithm by processing signals eight times faster and improving the SNR by 3.7 dB independently, and by 10.8 dB when combined with MA-MD [143]. Additionally, a nonlinear distortion correction algorithm can rectify charge coupled device (CCD) measurement spectra, eliminating errors that may arise during various Time of INTegration (TINT) measurements [142]. Furthermore, a fiber grating filter is incorporated to enhance the overall signal processing capabilities [144]. (3) In addition, advanced algorithms and models based on a large number of monitoring data are crucial for the real-time evaluation and prediction of battery SOC, SOH, and RUL in fiber optic sensing battery monitoring red[149,150,151,152,153,154]. This requires the combination of multi-dimensional parameters inside and outside the monitored battery with technologies such as machine learning and artificial intelligence to achieve a battery level battery management system (BMS) and fault diagnosis. Our recently proposed series of advanced fiber optic signal processing methods [155,156,157,158,159] is capable of eliminating various kinds of noises in practice, such as electrochemical noises, road noises, wind noises, and auxiliary system noises [160,161]. This helps in fiber optic sensing multiparameter decoupling; signal signature identification; and evaluation with battery SOC, SOH, and RUL.

Finally, the cost of equipment and standardization of production also limit the application of FBG sensors in battery monitoring. (1) Standardizing the integration of FBG sensors during the battery manufacturing process ensures their seamless incorporation into the battery structure. (2) The cost of optical sensing does not increase linearly with the number of batteries because of multiplexing, which is different from the cost of using electrical sensing. (3) The establishment of uniform standards and specifications is crucial to ensuring the applicability and reliability of different types of FBG sensors across various battery systems. According to a cost analysis forecast done by Alamgir in 2016 [162], fiber optic sensing BMSs are expected to compete with traditional electrical sensing BMSs.

In short, fiber optic sensing battery monitoring technology includes fiber optic design, fabrication, installation, signal processing, physical field analysis of monitoring objects, big data analysis, and artificial intelligence battery health assessment. Although there are still some difficulties and challenges, fiber optic sensors will play an important role in future development. The application of fiber optic sensors can not only guarantee the safe operation of batteries but also provide the necessary data support for the optimization and development of new high energy density batteries. Therefore, further research and application of fiber optic sensor technology is of great significance to promote the development of battery technology.

## 5. Discussion

This paper provides a comprehensive review and analysis of various sensing technologies, with a particular focus on the application of FBG technology in monitoring batteries. The strengths and limitations of different detection technologies are carefully examined. Assessing parameters crucial to battery health, current technologies can be categorized into:

(1) Traditional monitoring technologies, while accurate, face limitations in real-time monitoring due to the need for intricate sample preparation, complex signal processing, and expensive equipment.

(2) Fiber optic sensing technology, while capable of monitoring physical parameters related to battery status in real time, faces challenges in achieving accurate monitoring. Its effectiveness may vary under different conditions of use.

(3) Fiber optic sensing technology emerges as a promising avenue for monitoring chemical characteristics such as pH value, RI, and gas emissions in batteries. It holds the potential to overcome the drawbacks of the previous methods, enabling precise and real-time assessments of battery performance.

Therefore, the primary challenge in current battery monitoring technology lies in achieving real-time and accurate detection of SOH, SOC, and RUL. A promising approach involves the real-time monitoring of battery material properties, contingent upon establishing explicit monitoring parameters and physical models of the battery state. Despite facing challenges, such as standardized packaging, temperature–strain coupling for fiber optic sensing, noise reduction in signal processing, and the integration of multiparameter and artificial intelligence models, FBG technology is expected to comprehensively enhance the QRL of batteries. This optimism is grounded in the unique advantage of FBG, as evidenced by its pivotal role in advancing the field of battery condition monitoring [163,164,165].

## Figures and Tables

**Figure 1 sensors-24-02057-f001:**
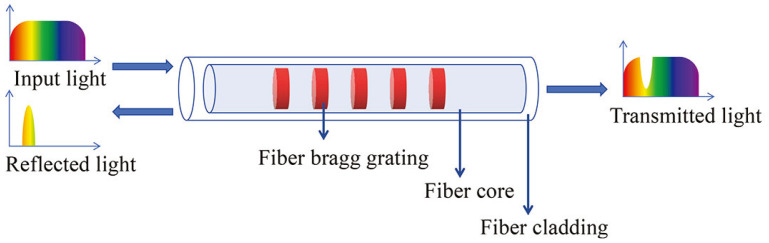
Typical scheme of FBG sensors.

**Figure 2 sensors-24-02057-f002:**
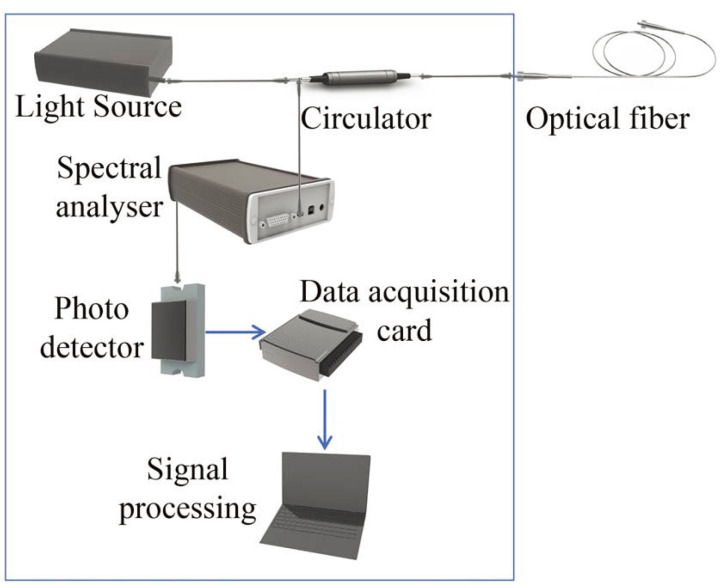
Description of the interrogation system.

**Figure 3 sensors-24-02057-f003:**
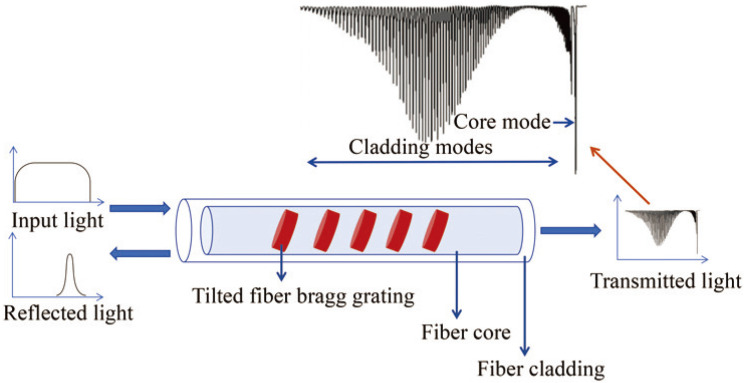
Typical scheme of TFBG sensors.

**Figure 4 sensors-24-02057-f004:**
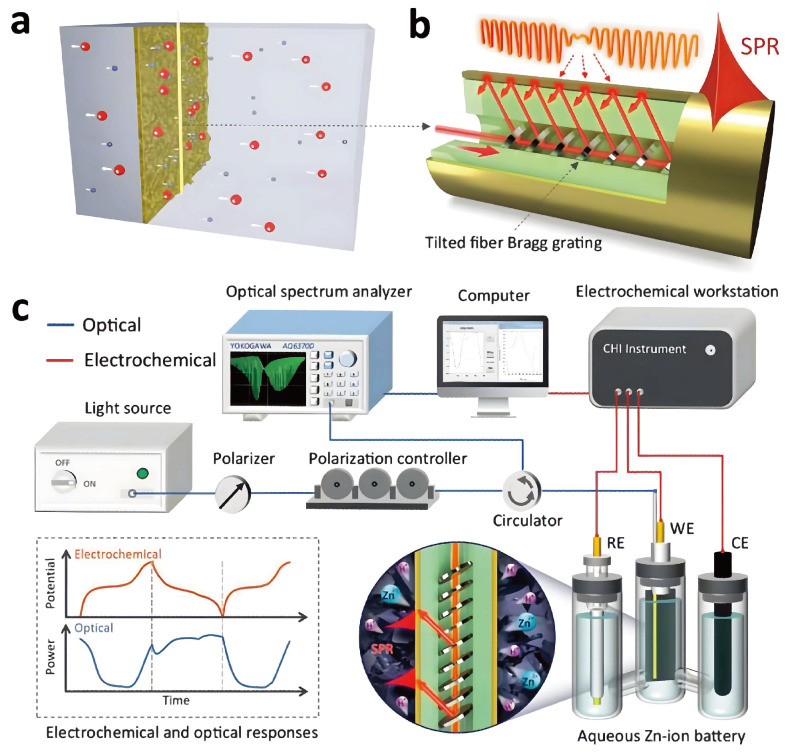
TFBG-based SPR optical fiber sensor [110]: (**a**) ionic concentration; (**b**) sketch of the configuration of a plasmonic fiber optic sensor; and (**c**) experimental setup for RI monitoring.

**Figure 5 sensors-24-02057-f005:**
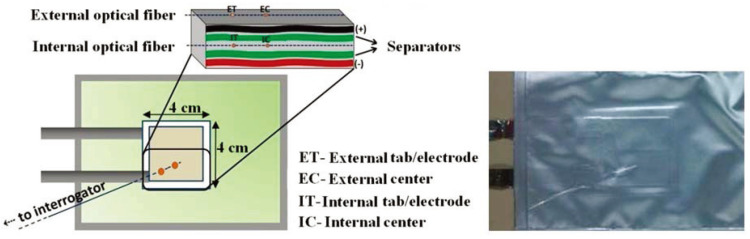
Monitoring temperature changes in pouch Li-ion battery using thermocouple sensors and FBG sensors [13].

**Figure 6 sensors-24-02057-f006:**
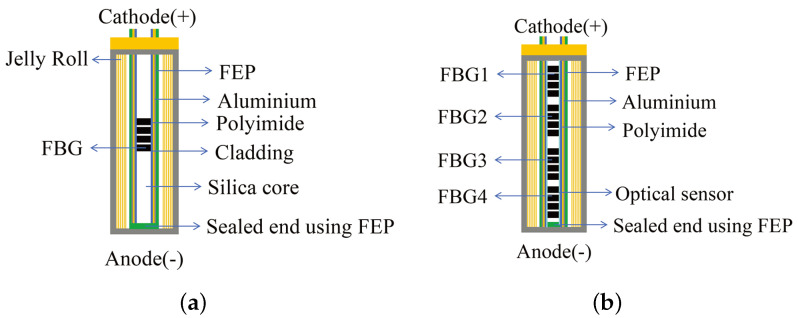
FBG-based temperature measurement for cylindrical Li-ion battery: (**a**) single FBG sensing element schematic [115]; (**b**) four FBG sensing elements uniformly distributed schematic [116].

**Figure 7 sensors-24-02057-f007:**
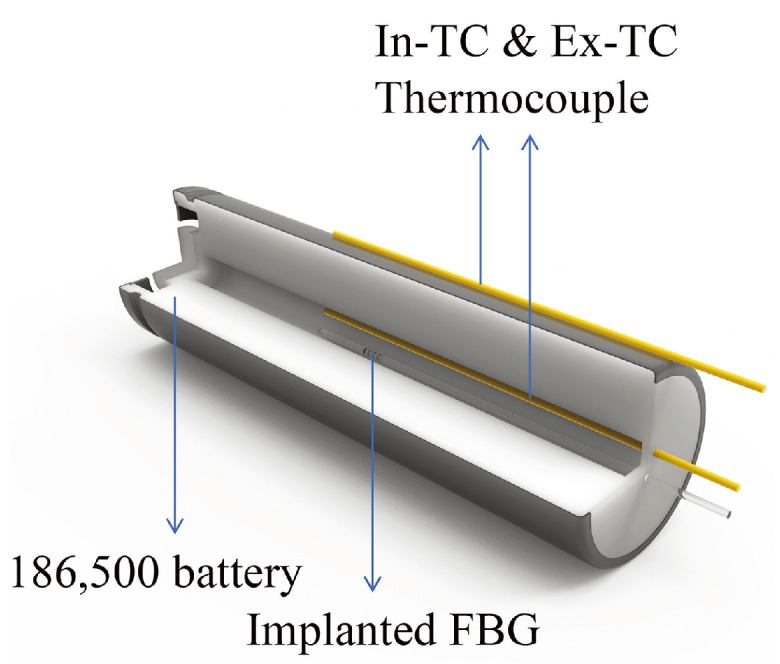
Schematic of integrating FBG sensors into the central void of cylindrical jelly-roll Li-ion battery [117].

**Figure 8 sensors-24-02057-f008:**
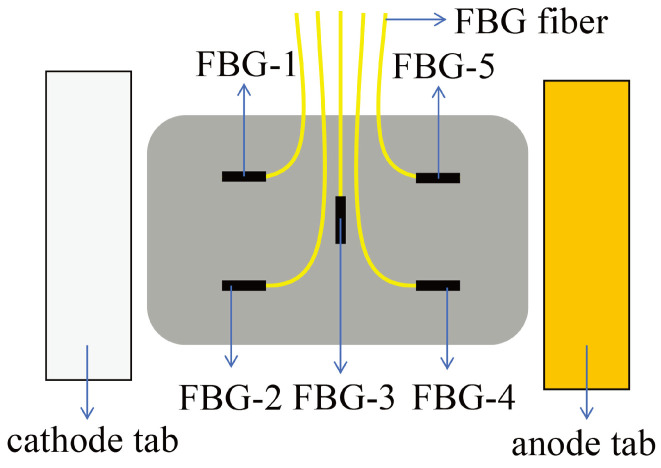
Schematic diagram of FBG array thin film temperature measurement device [15].

**Figure 9 sensors-24-02057-f009:**
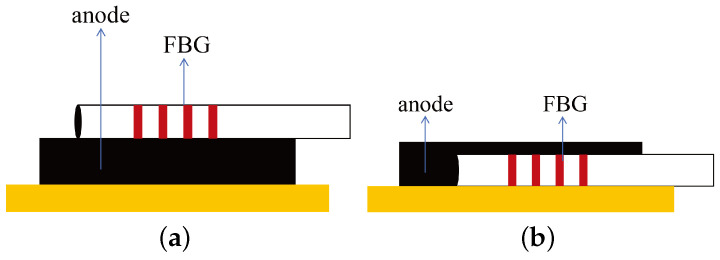
Two different approaches [118]: (**a**) internally attached FBG strain sensor on graphite anode; (**b**) internally implanted FBC strain sensor within graphite anode.

**Figure 10 sensors-24-02057-f010:**
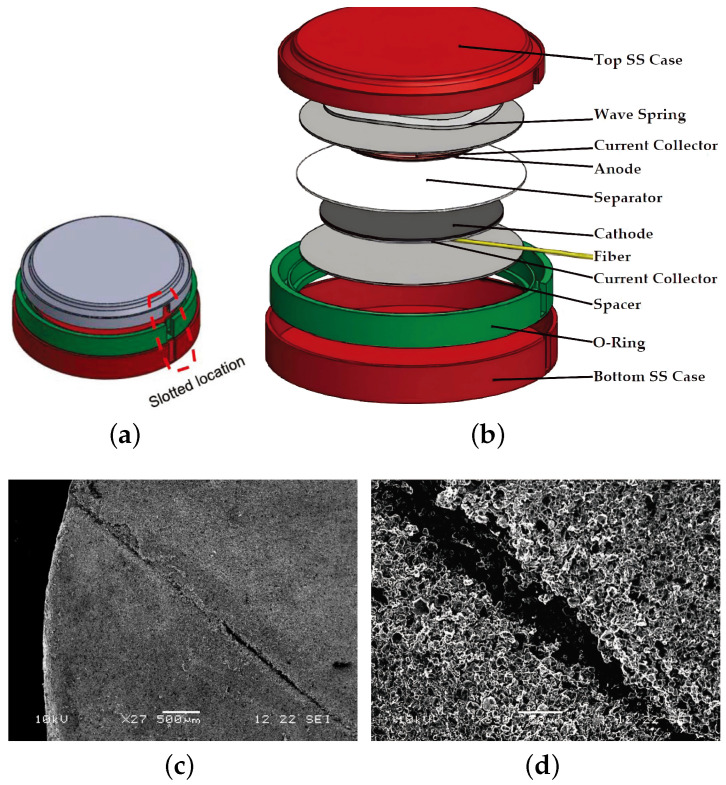
FBG sensor integrated into a Li-ion coin cell [24]: (**a**) case of the coin cell; (**b**) all components of the coin cell; (**c**) SEM image of the damaged cathode at 500 micron magnification; and (**d**) SEM image of the damaged cathode at 300 micron magnification.

**Figure 11 sensors-24-02057-f011:**
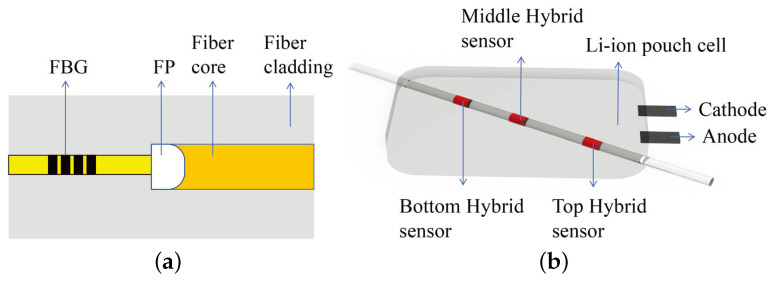
Simultaneous monitoring of temperature and strain scheme [122]: (**a**) FBG combined with FP; (**b**) diagram of the experimental setup of a sensor network for temperature and strain monitoring of Li-ion pouch battery.

**Figure 12 sensors-24-02057-f012:**
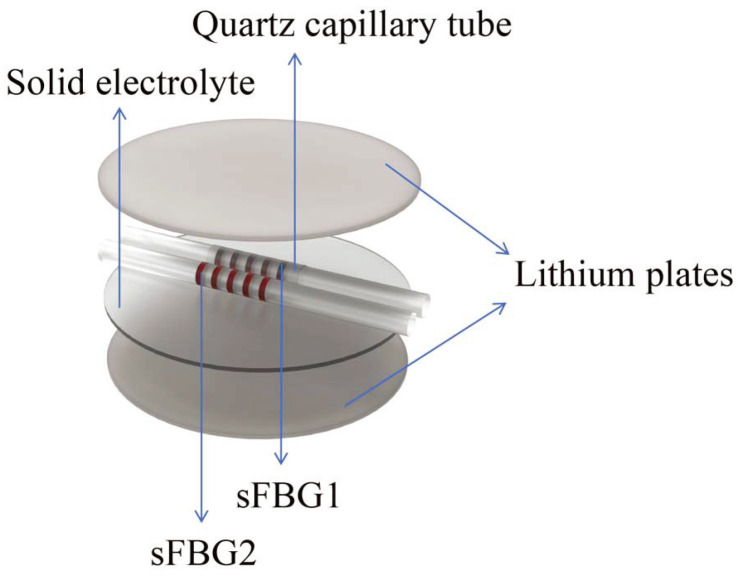
Schematic diagram of the location of the FBG sensors [112].

**Figure 13 sensors-24-02057-f013:**
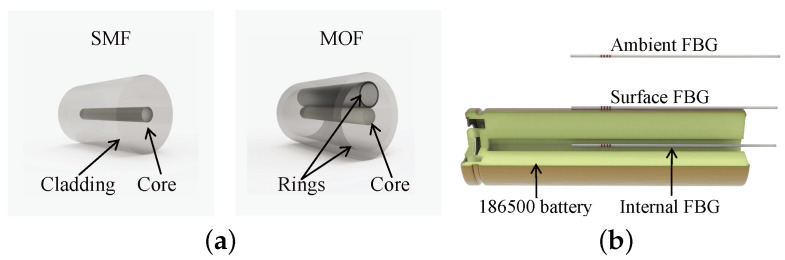
Monitor internal battery temperature and pressure: (**a**) cross-section of SMF and MOF; (**b**) SMF-FBG and MOF-FBG implanted into Li-ion battery.

**Figure 14 sensors-24-02057-f014:**
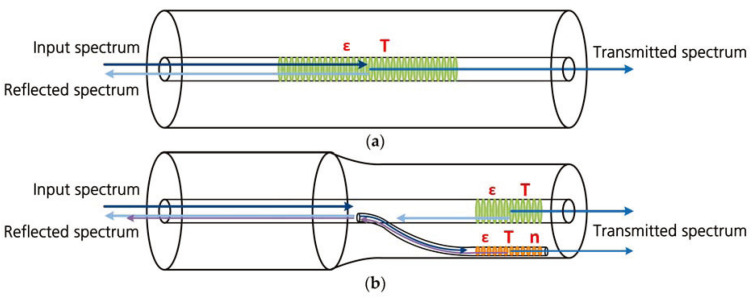
Schematic of typical and enhanced sensitivity optical waveguide structures [127]: (**a**) the original design, where the grating is inscribed in the fiber core; (**b**) the self-compensating design, where an additional waveguide is inscribed in the cladding to increase the refractive-index sensitivity and to direct some of the propagating light to this waveguide.

**Table 2 sensors-24-02057-t002:** Summary of FBG Sensor Monitoring Battery Performance.

Year	Battery Type	Measured Parameter	Accuracy	Sensitivity	Ref.
2016	Li-ion Pouch Battery	External and internal temperatures	0.1 °C	10.27 pm/°C	[13]
2018	Li-ion Cylindrical Battery	Internal temperatures	-	-	[115]
2018	Li-ion Cylindrical Battery	Internal temperatures	1 °C	11 pm/°C	[116]
2022	Li-ion Cylindrical Battery	Internal temperatures	0.5 °C	9.89 pm/°C	[117]
2023	Hardcase Li-ion Batteries	External and internal temperatures	0.5 °C	9.8 ± 0.2 pm/°C	[15]
2016	Li-ion Pouch Battery	Internal strain	-	-	[118]
2022	Swagelok Battery	Internal strain	-	-	[120]
2022	Li–S Pouch Battery	Internal strain	-	0.847 pm/με	[70]
2022	Li-ion Battery	Internal temperatures	-	11 ± 0.3 pm/°C	[121]
2017	Li-ion Coin Battery	Internal temperature and strain	-	-	[24]
2019	Li-ion Pouch Battery	Internal temperature and strain	0.1 °C, 0.1 με	40 pm/°C, 2.2 pm/με	[122]
2022	Li-ion Coin Battery	Internal temperature and strain	-	11.7 pm/°C, 11.3 pm/°C, 1.04 pm/με	[112]
2023	Li-ion Pouch Battery	Internal temperature and strain		10.3 pm/°C	[126]
2020	Li-ion Cylindrical Battery	Internal temperature and pressure	0.1 °C, 0.14 bar	−2.7 pm/bar (MOF), −0.3 pm/bar (SMF), 10 pm/°C (SMF), 10 pm/°C (MOF)	[124]
2021	Sodium-ion Cylindrical Battery	Internal temperatures and pressure	0.1 °C	−2.7 pm/bar (MOF), −0.3 pm/bar (SMF)	[125]
2019	Li-ion Pouch Battery	Electrolyte RI	-	-	[127]
2022	Aqueous Zn-ion Batteries	Electrolyte RI	-	-	[110]
2024	Lithium Metal Battery	Electrolyte RI	-	-	[111]
2021	Li-ion Cylindrical Battery	Internal temperature and electrolyte RI	6 × 10^−5^ RIU	−18 nm/RIU, 10.1 pm/°C	[90]
2016	Li-ion Pouch Battery	SOC	-	-	[119]
2018	Supercapacitors	SOC	-	-	[123]

## Data Availability

Not applicable.

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
