# Peer review of "Advancements in Battery Monitoring: Harnessing Fiber Grating Sensors for Enhanced Performance and Reliability"

_sensors, 2024, doi:10.3390/s24072057_

Round 1

Reviewer 1 Report

Comments and Suggestions for Authors

This paper highlights the advancements in battery monitoring technology, focusing on Fiber Bragg Grating (FBG). By delving into the causes of battery degradation and the principle of FBG, this study discusses key aspects of FBG sensing, including installation location, monitoring targets, and their correlation with optical signals. In addition, challenges in battery monitoring, such as standardizing of FBG packaging processes, decoupling multi-parameters, and controlling costs, are also addressed. However, in the abstract section a general overview is presented and I would suggest to include some quantitative information or data in the abstract section and manuscript structure as well. The introduction section could be extended as it’s a review paper. A detailed proofreading is also required for the manuscript such as a minor correction is required on sentence 148-149.

Comments on the Quality of English Language

Moderate editing of English language required.

Reviewer 2 Report

Comments and Suggestions for Authors

This review article presents the technological advancements in continuous battery monitoring with a focus on fiber Bragg gratings. Overall, the structure and organization of the article look good. I have several concerns that require additional clarification from the authors. Therefore, I recommend a minor revision.

1. Various fiber optic sensing techniques such as OFDR are available for strain and temperature measurements in a fully distributed manner. What motivated the selection of FBG sensors for battery monitoring? Incorporating insights from additional sources would be beneficial for the readers to better understand the choice (suggested article: 10.1109/JSEN.2020.3024943).

2. The citation in line 149 appears to be missing.

3. There is an error in Equation (1).

4. In line 177, the definitions of the parameters for Equation (1) seem to be confused. Furthermore, lines 177 and 186 contain redundant information.

5. While the language usage throughout the article is generally acceptable, I recommend a thorough review to correct grammatical mistakes, such as those found in line 221 and elsewhere.

Addressing these issues would improve the manuscript's clarity and depth, rendering it a more valuable addition to the field.

Comments on the Quality of English Language

Generally fine, but still has a few errors.

Reviewer 3 Report

Comments and Suggestions for Authors

I find the presented review interesting, relevant, and timely. It is logically constructed and well designed. I would recommend it for publication in Sensors after minor issues have been addressed:

1. What is meant by distributed measurement in Table 1? A fiber Bragg grating is a classic point-wise sensor. Or is it supposed to be interrogated by a coherent reflectometer or a frequency domain reflectometer?

2. In my opinion, basic information about interrogators needs to be added in section 2.2, since the spectrum or associated characteristics must be obtained and processed in some way.

3. Figure 7 appears in the middle of the paragraph, breaking the sentence into two parts. I suggest changing its position.

4. Do these studies always require an optical spectral analyzer or spectrometer? This is typically the most expensive element of the system, so for many applications researchers are moving away from wavelength measurements to intensity measurements by using slope filters. Does this apply to battery monitoring?

5. This question came up when I wrote comment 1. Is there a need to use distributed measurements when monitoring batteries? For example, for a grating arrangement as in Figure 5a, should the line be interrogated with an optical frequency domain reflectometer? If yes, then this can be discussed in the final sections of the article.

6. I propose to add the FDDA/ADFA methods for eliminating noise when processing the optical data [147-151]: [http://dx.doi.org/10.3390/a16090440], and also the following methods for processing signals from FBG: [https://doi.org/10.3390/s21082817], [https://doi.org/10.3390/app11178189].

Comments on the Quality of English Language

Minor editing of English language is required.
